# Should We Screen HIV-Positive Migrants for Strongyloidiasis?

**DOI:** 10.3390/pathogens9050388

**Published:** 2020-05-18

**Authors:** Caroline Theunissen, Emmanuel Bottieau, Marjan Van Esbroeck, Achilleas Tsoumanis, Eric Florence

**Affiliations:** Institute of Tropical Medicine, Department of Clinical Sciences, 2000 Antwerp, Belgium; ebottieau@itg.be (E.B.); mvesbroeck@itg.be (M.V.E.); atsoumanis@itg.be (A.T.); eflorence@itg.be (E.F.)

**Keywords:** strongyloidiasis, HIV infection, migrants

## Abstract

Background: *Strongyloides stercoralis,* a nematode endemic in all (sub)tropical regions, can cause life-threatening disease, especially in immunosuppressed patients. Many countries with high HIV-prevalence rates are also highly *S. stercoralis* endemic, and co-infection may occur. Methods: Retrospective study based on serological testing for *S. stercoralis* in all HIV-infected migrants followed at the Institute of Tropical Medicine, Antwerp, Belgium. If untested, serologic testing was performed on stored samples, dating from the first HIV viral load determination. The epidemiological, clinical and laboratory features of patients with and without strongyloidiasis were retrieved from the electronic medical files. Results: Of the 2846 HIV patients in active follow-up, 723 (25.4%) had a migration background. Thirty-six patients (5.1%) were diagnosed with *Strongyloides* co-infection, including 29 during their medical evaluation and seven retrospectively. Patients had a median age of 35.3 years (IQR 30.3–44.4), 28 patients (78%) originated from Sub-Saharan Africa and median time in Belgium was 3.5 years (IQR 0.8–5.7). Symptoms compatible with strongyloidiasis were present in 17 (47%) patients, of whom two were diagnosed retrospectively. Eosinophilia (eosinophil cell count > 450/µL) was observed in 19 (53%) participants. Median CD4 level was 386 /µL (IQR 299–518) at diagnosis of co-infection. Of note, 8 (22%) patients with strongyloidiasis had no reported symptoms nor eosinophilia. None of the patients developed hyperinfection syndrome. There were no differences in age, gender, geographic origin, clinical presentation, CD4 level or viral load between patients with and without strongyloidiasis. Only eosinophilia was strongly correlated with the presence of *Strongyloides* in multivariate analysis (OR 10.74 (95% CI 5.19–22.25), *p* < 0.001); the positive likelihood ratio (LR+) of eosinophilia for strongyloidiasis was 5.38 (95% CI 3.66–7.91). Conclusion: Strongyloidiasis was diagnosed in 5.1% of HIV-infected migrants. Eosinophilia had good confirming power for the presence of the disease. However, a sizeable proportion (22%) of co-infected individuals were asymptomatic and had normal eosinophil count, supporting universal screening of all HIV-positive patients native to tropical countries.

## 1. Introduction

Strongyloidiasis is a neglected tropical disease with an estimated globally 350 million infected individuals [1]. The infection is caused by *Strongyloides stercoralis*, a nematode characterised by the unique feature of auto-infection, i.e., the capacity to replicate indefinitely within the human host and cause lifelong infection. The disease is endemic in tropical and subtropical regions and parts of the Southeastern United States [2]. Clinical presentation varies from asymptomatic carriership to life-threatening disease (hyperinfection syndrome and disseminated strongyloidiasis). Risk factors for severe disease include certain immunosuppressive conditions, such as HTLV1-infection, treatment with corticosteroids, solid-organ and bone-marrow transplantation or alcoholism [3,4]. Serologic testing is the current gold standard for diagnosis but lacks sensitivity in certain conditions, such as acute infections and immunosuppressed patients, and specificity in case of concomitant nematode infections [5,6,7]. The recommended treatment of non-complicated disease is a single course of ivermectin (0.2 mg/kg), a highly effective, well-tolerated and affordable drug [8]. 

The current Centers of Disease Control (CDC) guidelines recommend screening for strongyloidiasis in persons who have traveled to an endemic region, especially if they are or going to be immunosuppressed [9]. Guidelines set by the European CDC provide similar recommendations and suggest screening in migrants originating from high-endemic countries in order to reduce individual morbidity through early detection and treatment [10,11]. Although persons with HIV/AIDS can have disseminated strongyloidiasis or hyperinfection syndrome, observational studies have not shown an increased risk of such complications in this specific population, in contrast to other immunosuppressive conditions [12]. However, HIV and AIDS are often associated with other risk factors for severe strongyloidiasis, such as opportunistic infections requiring steroid treatment or malignancies. 

The main aim of this study was to determine the frequency of strongyloidiasis in a cohort of HIV-positive migrants in active follow-up at the HIV clinic of the Institute of Tropical Medicine, Antwerp (ITMA), Belgium. Secondary objectives were to describe the clinical and laboratory presentation, treatment and outcome of patients found with *Strongyloides*-HIV co-infection and to identify epidemiological, clinical and laboratory predictors of co-infection.

## 2. Results

Of the 2846 HIV patients in active follow-up at ITMA, 723 (25.4%) met the inclusion criteria (Figure 1): 518 patients (72%) originated from Sub-Saharan Africa, 95 (13%) from South and Central America, 87 (12%) from Asia and 23 (3%) from North Africa. The top five Sub-Saharan origin countries were Cameroon, followed by the Democratic Republic of the Congo, Nigeria, Rwanda and Burundi. The top three countries of origin in South and Central America were Brazil, Suriname and Ecuador. More than half of the Asian migrants originated from Southeast Asia, and the majority (35 patients) came from Thailand. From the 23 North African patients, 16 were from Morocco.

### 2.1. Frequency of Strongyloidiasis (Figure 1)

*Strongyloides* serologic testing was already performed in 393/723 (54%) HIV-infected migrants and was positive in 29 of them (7.3%). A stored sample for retrospective testing was available in 317 (93.5%) of the remaining 330 patients. Serology yielded a positive result in seven of them (2.2%). Overall frequency of strongyloidiasis in our cohort of HIV-infected migrants was 5.07% (95% CI: 3.68–6.94) (36 patients/710 tested). Seroprevalence was 5.81% (95% CI: 2.51–12.90) in Asians; 5.50% (95% CI: 3.83–7.84) in Sub-Saharan Africans; 3.22% (95% CI: 1.10–9.06) in Central and South Americans and 0% (95% CI: 0–14.87) in North Africans.

### 2.2. Clinical Presentation

The demographic, clinical and laboratory parameters of HIV positive migrants with and without strongyloidiasis are illustrated in Table 1.

Symptoms compatible with strongyloidiasis were present in 17 (47%) of the 36 patients with a positive *Strongyloides* serology. The most frequent symptom was abdominal pain (19%), followed by cough (17%), itching (8%) and skin rash (3%). None of the patients had diarrhoea. Two of the symptomatic cases were diagnosed retrospectively during this study. None of the patients developed hyperinfection syndrome or disseminated strongyloidiasis during their follow-up. Eosinophilia was present in 19 patients (53%) with strongyloidiasis. Of note, eight (22%) patients had no symptoms nor eosinophilia at the time of diagnosis. Stool microscopy and PCR for *Strongyloides* were performed, respectively, in 18 (positive in three) and 5 (positive in two) patients.

Six patients with strongyloidiasis had at least one concomitant helminthic infection: five had *Schistosoma* antibodies (on 21 tested) and four had *Filaria* antibodies (on 10 tested and including one with *Mansonella perstans* microfilaria in blood), one with *Echinococcus granulosus* antibodies (on one tested) and one with *Ankylostoma* eggs detected in the faeces. 

Twenty-two patients received variable courses of ivermectin treatment: a single dose in 14, a two-day course in 4, and a three-day course in another 4 patients. Thirteen patients were not treated for the following reasons: presence of an alternative diagnosis in three, negative Strongyloides serologic testing at a following visit in three and no clear reason in the remaining eight patients. Treatment for strongyloidiasis was administered in 14 out of the 19 patients with eosinophilia, after which the eosinophil count normalised in 10 of them. In 10 patients, a follow-up serology was performed, of which eight became negative and two remained positive at resp. 10 and 18 months post-treatment.

There were no differences in age, gender, geographic origin, clinical presentation, CD4 level or viral load between HIV patients with and without strongyloidiasis (Table 1). In bivariate analysis, patients with “any symptom” tended to be more often diagnosed with strongyloidiasis, while association with eosinophilia was very strong. Only the presence of eosinophilia remained strongly correlated with *Strongyloides* infection in multivariate analysis (OR 10.74 (95% CI 5.19–22.25), *p* < 0.001); the positive likelihood ratio (LR+) of eosinophilia for strongyloidiasis was 5.38 (95% CI 3.66–7.91).

## 3. Discussion

The prevalence of strongyloidiasis in the cohort of HIV-infected migrants seen at the ITMA, Belgium was 5.1% and varied little across regions of origin. In our experience, about 20% of the cases had been missed by the treating HIV physician during initial evaluation and routine follow-up. No symptom was clearly associated with the diagnosis of strongyloidiasis, but eosinophilia appeared to have good confirming power for the presence of the disease in this population. 

Despite the retrospective character of our study, we managed to include and perform serology on almost all actively followed-up patients at our clinic, thanks to the availability of stored plasma samples. This resulted in a large, nearly complete and therefore representative cohort of 710 HIV-infected migrants. Our 5.1%-seroprevalence is, however, lower when compared to similar migrant HIV-positive populations from other Western countries, where rates of up to 26% have been reported [13]. In general, studies of strongyloidiasis in HIV-infected migrants, although scarce and based on relatively small patient numbers, report highly variable prevalence rates. Globally, prevalences of *Strongyloides* infection in HIV-positive individuals (irrespective of their migration status) and in migrants (irrespective of their HIV status) were estimated at respectively 10% and 12.2% in two recent meta-analyses, but the included studies showed significant heterogeneity in the diagnostic approach of strongyloidiasis and were therefore difficult to compare [14,15]. Concerning strongyloidiasis in HIV-infected migrants, one prospective study of a U.S. AIDS cohort describes prevalence of 25% but relates this high percentage to a possible low specificity of the used enzyme-linked immunosorbent assay using crude *S. stercoralis* extract [16]. Two retrospective studies in HIV-positive migrants, one in Italy [17] and one in Canada [18], report prevalence rates of around 10–11%, but in the former, four out of the 15 patients with strongyloidiasis were diagnosed by stool microscopy since their serologic testing result was negative. In general, stool microscopy and PCR are considered less sensitive than serology in the diagnosis of strongyloidiasis. However, in our study, stool microscopy and PCR were performed in only a limited amount of patients, and therefore we cannot draw any conclusion on the sensitivity of this diagnostic method in our study population. Although serology is considered the most sensitive method to diagnose *Strongyloides* infection in migrants [11], there are two reasons why our study may underestimate the calculated frequency of strongyloidiasis. First, it is possible that some serological tests were false-negative, especially in patients with very low CD4 counts. The diagnostic accuracy of *Strongyloides* serology in HIV patients has, to our knowledge, not been robustly studied. Second, serology was performed retrospectively, meaning nearly half of all cases involved stored samples. We cannot exclude antibody degradation, induced by prolonged preservation, although most of these samples had never been thawed before. Moreover, we have never experienced antibody degradation of stored samples in our setting before. On the other hand, our seroprevalence rate could also be somehow overestimated since false-positive results can occur with other nematode infections, either documented (6 out of 36 patients; 19% in our study) or not [19]. In general, however, this cross-sectional study provides robust data on the prevalence of strongyloidiasis in a large proportion of well-characterised HIV-positive migrants in Belgium, attended by a stable team of experts in HIV and tropical diseases. 

The geographic distribution of strongyloidiasis in our HIV-positive migrant population showed that most cases originated from Sub-Saharan Africa, followed by Southern Asia and Latin America. It is, however, noteworthy that prevalences were rather similar in these three subgroups of migrants, while no case was observed in patients originating from North Africa. In general, frequencies observed here were much lower than certain prevalence rates reported in persons living in endemic countries, although these rates can vary considerably depending on the diagnostic method used and the studied population and/or region: up to 35% in Sub-Saharan Africa [20], and up to 24% in community-based surveys in Thailand (17.9% in HIV-infected individuals) [15,21]. In the non-endemic setting, strongyloidiasis is present in migrant populations from any tropical region with no clear differences between HIV-positive and HIV-negative individuals [15,22]. 

In contrast to a former report, describing a higher frequency of diarrhoea, cough or skin disorders in co-infected patients [17], we could not detect any discriminative epidemiological or clinical feature for strongyloidiasis in our HIV population. Nor could any association be found between the presence of the disease and CD4 count and/or viral load, as was the case in two other cross-sectional studies [13,23]. The lack of predictive symptoms might be partly explained by the retrospective character of the study, limiting the accurate capture and reporting of symptoms related to strongyloidiasis. In addition, most participants were screened at the time of HIV diagnosis, complicating the correct attribution of the symptoms, in particular for a disease with non-specific features in an HIV population also vulnerable to many other infections. However, other studies also demonstrated that nonspecific symptoms such as abdominal pain, diarrhoea or skin manifestations alone were not always associated with the diagnosis of strongyloidiasis in patients with and without HIV [13,16,24]. While the clinical presentation was little helpful, the presence of eosinophilia had good confirming power for strongyloidiasis in our population of HIV-positive migrants (positive LR of 5.4) [25]. Of note, some authors reported a reduced level of eosinophil counts in HIV–*Strongyloides* co-infected patients, when compared to other immunosuppressed patients or to immunocompetent individuals with strongyloidiasis. One of the reasons might be the inhibitory role of the HIV virus on the production of eosinophils [26]. 

Although observational studies have not shown an increased risk of severe strongyloidiasis in HIV-infected patients, the complex reciprocal link between the parasite and HIV remains to be fully elucidated [12]. HIV infection has been associated with a two-times-higher risk of acquiring strongyloidiasis, and cases of strongyloidiasis-immune restitution inflammatory syndrome (IRIS) have been described, appearing after starting both highly active antiretroviral therapy (HAART) and anti-helminthic therapy [14,27,28]. Moreover, chronic immune activation induced by parasite infestation has been suggested as one factor that adversely influences epidemics of HIV/AIDS in Africa [29]. In addition, since the high effectiveness of current HAART has dramatically improved the life expectancy of HIV-infected individuals, immunosuppressive agents for any kind of condition are increasingly administered to them in the same way as the general population [23]. This is why we advocate systemic screening for strongyloidiasis in HIV patients with a migration background and a history of possible exposure to the parasite, in order to treat early and prevent long term morbidity and mortality in this population. 

## 4. Materials and Methods 

### 4.1. Patients

All patients with a migration background and actively followed up at the reference HIV clinic of the ITMA were included. Migration background was defined as originating from a *Strongyloides*-endemic region before emigrating to Belgium, including South and Central America, North Africa, Sub-Saharan Africa and Southern Asia. Active follow-up at the ITMA HIV clinic was defined as having had a least 2 visits during the course of the follow-up and a last visit no longer than one year before 1 January 2018. Strongyloidiasis was defined as having a positive serology for *S. stercoralis* (see definition below). Results of serological testing for *S. stercoralis* were retrieved from the laboratory database. In case serologic testing had never been performed during follow-up, it was done retrospectively for the purpose of this study on a stored plasma sample, dating from the patient’s first HIV viral load (VL) determination. The epidemiological, clinical and laboratory features of all patients were retrieved in the patient files and further analyzed. Symptoms compatible with strongyloidiasis were categorised in abdominal (abdominal pain, diarrhoea, anorexia/weight loss), cutaneous (itching, skin rash) and respiratory (cough, dyspnea, chest pain) symptoms. Eosinophilia was defined as a blood eosinophil level above 450/µL.

### 4.2. Serology

Detection of *Strongyloides* antibodies during routine follow-up was performed on serum samples at the medical laboratory of ITMA by means of the *Strongyloides* Serology Microwell ELISA kit (IVD Research Inc. Carlsbad, CA, USA) detecting IgG antibodies against *S. stercoralis* antigen until October 2009 and by means of the *Strongyloides ratti* ELISA (Bordier Affinity Products, Crissier, Switzerland), in which specific IgG reacts with *Strongyloides ratti* somatic larval antigens thereafter. For the HIV-positive migrants who had not yet been tested, *Strongyloides* serology was performed on stored samples of the first HIV VL determination, stored at −80 °C since 1996, using the *Strongyloides ratti* ELISA (Bordier Affinity Products). Both kits were used according to the instructions of the manufacturer. Results from both tests were considered positive in case of an optical density ratio ≥ 1.0. The medical laboratory is accredited for *Strongyloides* serology according to ISO15189 as of 2010.

### 4.3. Statistical Analysis

The frequency of strongyloidiasis was estimated with 95% Wilson confidence interval. Categorical variables were expressed as absolute frequencies and percentages of the whole migrant HIV cohort. Quantitative variables were expressed as median and interquartile range (IQR). For the identification of the predictors of co-infection, patients were distributed in 2 groups: HIV patients with and without strongyloidiasis. Numeric parameters were compared using the Mann–Whitney U test and categorical variables using the Pearson chi-squared test or Fisher’s exact test. Results were described as risk ratio (RR) or as difference in medians, with 95% confidence interval (CI). A multivariate analysis was performed using logistic regression. The inclusion of variables in the model was decided according to their significance in the univariate analysis, together with the clinical importance of each variable regardless of whether or not it was significant in this analysis. All *p*-values <0.05 were considered as statistically significant. For the significant covariates, positive likelihood ratios (LR+) were calculated with 95% CI. The R version 3.5.1 software was used for statistical analysis.

The study protocol has been approved by the IRB of the ITMA and all patients gave consent for secondary use of their clinical data and laboratory results. The EC approval code of our study is EC/UZA/1815203.

## 5. Conclusions

Strongyloidiasis was diagnosed in 5.1% of HIV-infected migrants, regardless of the continent of origin, and was only positively correlated with the presence of eosinophilia, supporting a low threshold for screening in clinical care. The absence of symptoms and the normal eosinophil count in a sizeable proportion (22%) of HIV-positive individuals with strongyloidiasis could even justify a systematic screening since the disease can have a deleterious long-term impact, particularly in immunosuppressed patients. Well-designed cost-effectiveness studies are, however, highly needed to fully backup such general policy recommendations.

## Figures and Tables

**Figure 1 pathogens-09-00388-f001:**
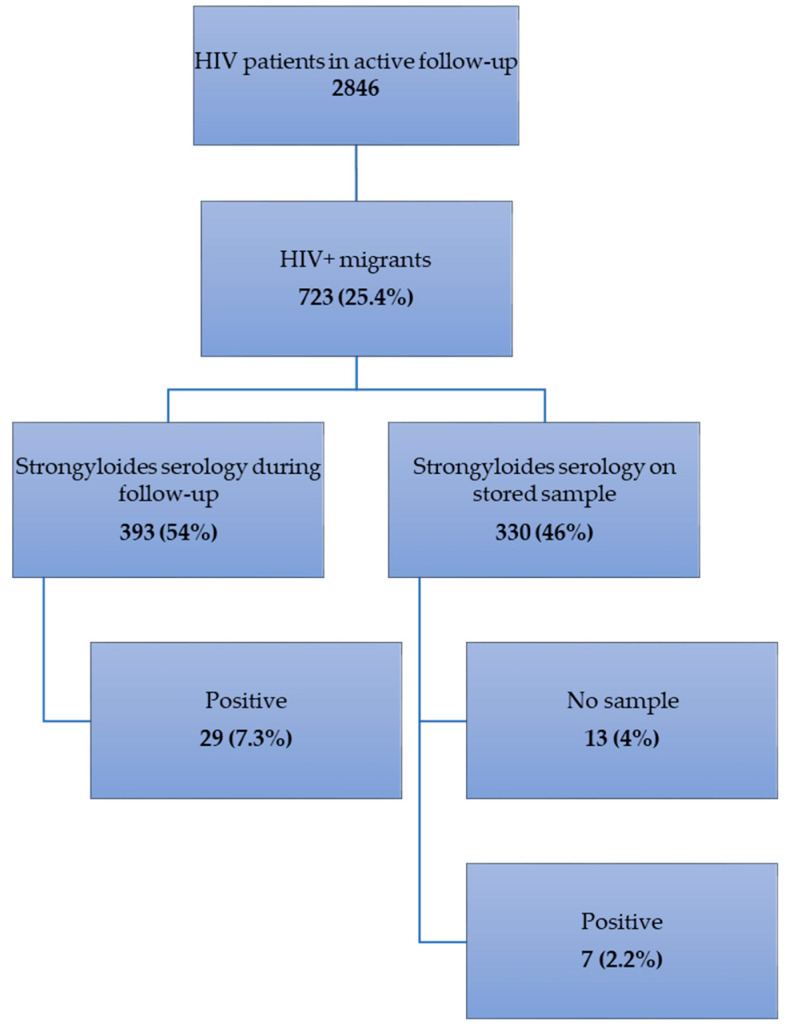
Frequency of strongyloidiasis in 710 HIV-infected migrants at ITMA, Belgium.

**Table 1 pathogens-09-00388-t001:** Demographic, clinical and laboratory parameters in 710 HIV infected migrants with and without strongyloidiasis.

Parameters	With Strongyloidiasis	Without Strongyloidiasis	*p*-Value
	(n = 36)	(n = 674)	
Serologic testing on doctor’s request (n, %)	29 (81)	364 (54)	<0.001
Age (y) (median, IQR)	35.3 (30.3–44.4)	34.6 (29.4–40.9)	0.39
Male gender (n,%)	19 (53)	300 (45)	0.42
Origin (n, %)			0.71
Sub-Saharan Africa	28 (78)	480 (71)	
Asia	5 (14)	81 (12)	
South & Central America	3 (8)	90 (14)	
North Africa	0 (0)	22 (3)	
Time in Belgium(y) (median, IQR)	3.5 (0.8–5.7)	2.1 (0.9–6.4)	1.00
Mode of transmission (n, %)			0.42
heterosexual	26 (72)	458 (68)	
Homo- and bisexual	9 (25)	142 (21)	
other	1 (3)	28 (4)	
unknown	0 (0)	46 (7)	
CD4 (median, IQR)	386 (299–518)	374 (234–531)	0.53
CD4 < 100 (n, %)	2 (6)	57 (8)	0.76
Undetectable VL (n, %)	5 (14)	185 (27)	0.08
Symptoms (n, %)			
Any	17 (47)	214 (31)	0.07
Abdominal	10 (28)	96 (16)	0.1
pain	7 (19)	39 (6.5)	
diarrhoea	0	25 (4)	
anorexia/weight loss	3 (8)	17 (2.5)	
Cutaneous	3 (8)	66 (11)	0.79
rash	1 (3)	45 (7.5)	
itching	3 (8)	29 (5)	
Respiratory	7 (19)	55 (9)	0.07
chest pain	1 (3)	14 (2)	
cough	6 (17)	38 (6)	
dyspnoea	0	3 (0.5)	
Eosinophils/µL (median, IQR)	555 (162.5–1167.5)	120 (50–230)	<0.001
Eosinophilia > 450 (N, %)	19 (53)	66 (10)	<0.001

IQR: interquartile range.

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
