# Peer review of "Should We Screen HIV-Positive Migrants for Strongyloidiasis?"

_pathogens, 2020, doi:10.3390/pathogens9050388_

Round 1

Reviewer 1 Report

Theunissen and coworkers present a retrospective serologic study for S. stercoralis infection in HIV-infected migrants and found strongyloidiasis in 5.1% of HIV-infected migrants.  The study was well-done, but there is one major uncertainty in the methodology, such as the sensitivity of the serologic test in HIV patients, esp. those with low CD4 counts.  The authors admit this, but it is a problem with the study.

I have only a few comments to improve the manuscript.    

lines 14, 44, 72, 74, 88, 104, 119, 138, 143, 209: use Serologic testing, not serology

line 88, in regard to stool microscopy, how many samples were examined?  Stool ova and parasite exams are very insensitive for the diagnosis of strongyloidiasis (see: Nielsen, P.B.; Mojon, M. Improved diagnosis of Strongyloides stercoralis by seven consecutive stool specimens. Zentralbl. Bakteriol. Mikrobiol. Hyg. A 1987263, 616–618

Author Response

1.Theunissen and coworkers present a retrospective serologic study for S. stercoralis infection in HIV-infected migrants and found strongyloidiasis in 5.1% of HIV-infected migrants.  The study was well-done, but there is one major uncertainty in the methodology, such as the sensitivity of the serologic test in HIV patients, esp. those with low CD4 counts.  The authors admit this, but it is a problem with the study.

Answer: We fully acknowledge the fact that the sensitivity of serologic testing for strongyloidiasis in HIV patients has not been thoroughly studied yet and we therefore mention this problem as a major caveat in the beginning of the discussion. This lack of sensitivity can indeed underestimate the prevalence of the disease in our study population.

2. lines 14, 44, 72, 74, 88, 104, 119, 138, 143, 209: use Serologic testing, not serology

Answer: the manuscript has been adapted accordingly except for lines 72, 104, 119 and 138 in which the word serology is not present. In stead we changed serology in lines 90 and 107 into serologic testing.

3. line 88, in regard to stool microscopy, how many samples were examined?  Stool ova and parasite exams are very insensitive for the diagnosis of strongyloidiasis (see: Nielsen, P.B.; Mojon, M. Improved diagnosis of Strongyloides stercoralis by seven consecutive stool specimens. Zentralbl. Bakteriol. Mikrobiol. Hyg. A 1987263, 616–618

Answer: We again acknowledge the poor sensitivity of stool microscopy for diagnosing strongyloidiasis although this method has been proposed and recommended as an additional diagnostic tool in immunosuppressed patients (Requena-Méndez A, Buonfrate D, Gomez-Junyent J, Zammarchi L, Bisoffi Z, Muñoz J. Evidence-Based Guidelines for Screening and Management of Strongyloidiasis in Non-Endemic Countries. Am J Trop Med Hyg. 2017 Sep;97(3):645-652.). In our retrospective study, stool microscopy was performed in 45 of the entire 723 HIV positive migrants and in 18 of those with a positive serology for strongyloides. Three of them showed larvae of S stercoralis (= lines 98-99).

Reviewer 2 Report

This is an interesting paper, making an important contribution to our understanding of the relationship between HIV and strongyloidiasis, an area that is poorly understood.

In general, the study is well thought out, and clearly described in the manuscript.

Two things need to be explained more clearly in the manuscript: the relationship between a positive ELISA IgG test for strongyloides and past infection, and eosinophilia as a predictor of strongyloidiasis.

Our present understanding is that a positive ELISA test such as you have used very likely indicates a current infection. I have explained this in more detail below.

Even though there is a highly significant difference in eosinophilia between strongylodes-positive and strongyloides-negative patients, eosinophilia has a low positive predictive value in your group, so is a poor predictor of strongyloidiasis. This is also borne our in the literature. I have explained this in more detail below.

Abstract:

Line 21 and 22, says that 47% of patients had symptoms compatible with strongyloidiasis, but doesn’t say that (line 106 and Table 1) that there was no significant difference in clinical presentation in patients with and without strongyloidiasis.

Line 47 Comment: positive serology and previous infection. It is unlikely that serology is positive because of a previous infection. Two of your references do not give a reference for their statement, and the third said that it is an open question. In the earliest reports that suggested that serology may be indicating previous infection, the efficacy of the treatment was unknown and researchers were relying on negative faecal testing as a criterion for cure. Kearns et al

found that seroprevalence of Strongyloides fell from initially 21% (n=818) before treatment with ivermectin to 5% six months after treatment in the first year of their study, and obtained similar results with a new group in the second year of the study. (Kearns TM, Currie BJ, Cheng AC, McCarthy J, Carapetis JR, Holt DC, et al. (2017) Strongyloides seroprevalence before and after an ivermectin mass drug administration in a remote Australian Aboriginal community. PLoS Negl Trop Dis 11(5): e0005607. https://doi.org/10.1371/journal.pntd.0005607

Line 115 Delete “and was rather” (evidently not deleted from an earlier version of the manuscript).

Line 118 Comment: If you relied on eosinophilia as a predictor of strongyloidiasis, nearly half of the infections would be missed. The positive predictive value of eosinophilia in your group is 19/(19+66) which is poor.

Line 173 I agree with your statement here that eosinophilia had a good confirming power for strongyloidiasis in your population.

Line 241 Comment: In Table 1, 53% had eosinophilia, so surely 47% had a normal eosinophil count.

Line 239 After “eosinophilia”, add “in 53% of the patients”

Author Response

1. Line 21 and 22, says that 47% of patients had symptoms compatible with strongyloidiasis, but doesn’t say that (line 106 and Table 1) that there was no significant difference in clinical presentation in patients with and without strongyloidiasis.

Answer: This is stated a bit further in the abstract (line 26): "There were no differences in age, gender, geographic origin, clinical presentation, CD4 level or viral load between patients with and without strongyloidiasis".

2. Line 47 Comment: positive serology and previous infection. It is unlikely that serology is positive because of a previous infection. Two of your references do not give a reference for their statement, and the third said that it is an open question. In the earliest reports that suggested that serology may be indicating previous infection, the efficacy of the treatment was unknown and researchers were relying on negative faecal testing as a criterion for cure. Kearns et al found that seroprevalence of Strongyloides fell from initially 21% (n=818) before treatment with ivermectin to 5% six months after treatment in the first year of their study, and obtained similar results with a new group in the second year of the study. (Kearns TM, Currie BJ, Cheng AC, McCarthy J, Carapetis JR, Holt DC, et al. (2017) Strongyloides seroprevalence before and after an ivermectin mass drug administration in a remote Australian Aboriginal community. PLoS Negl Trop Dis 11(5): e0005607. https://doi.org/10.1371/journal.pntd.0005607

Answer: It is true that Sudarshi and Luvira's conclusions on the specificity of serologic testing are based on the absence of symptoms and/or larvae in stool (which are both indeed no exclusion criteria for infection), and that the treatment modalities were not very clear in both papers. On the other hand, serology (optic density or OD) can stay positive for several months or even years, despite adequate treatment, reason why some authors propose to use an OD ratio based on pre- and post-treatment values to determine treatment response. Kearns et al indeed show a significant decline in serology results after a 12-months-interval MDA with ivermectin in an Aboriginal community, but on the other hand also show a seroreversion failure rate in 18% in a subgroup of their study population, indicating persisting antibodies and/or reinfection. In order to avoid controversy however, we propose to withdraw "or previous infection" from the manuscript, if the reviewer and editor agree

3. Line 115 Delete “and was rather” (evidently not deleted from an earlier version of the manuscript).

Answer: has been removed from the manuscript

4. Line 118 Comment: If you relied on eosinophilia as a predictor of strongyloidiasis, nearly half of the infections would be missed. The positive predictive value of eosinophilia in your group is 19/(19+66) which is poor.

Answer: We entirely agree with the reviewer's remark on this wrong formulaton in our manuscript. Therefore we replaced "predicted" and "predictor" by "positively correlated" and/or "good confirmation power" where appropriate (Lines 30-31, 130 and 256)

5. Line 173 I agree with your statement here that eosinophilia had a good confirming power for strongyloidiasis in your population.

Answer: see 4.

6. Line 241 Comment: In Table 1, 53% had eosinophilia, so surely 47% had a normal eosinophil count.

Answer: I am afraid I don't really understand what you mean. In our version of the manuscript this sentence does not appear in Line 241 (or elsewhere). 

7. Line 239 After “eosinophilia”, add “in 53% of the patients”

Answer: In our manuscript the word "eosinophilia does not appear in Line 239, please provide more details so that I can adapt the manuscript accordingly.